# Why Are X Autosome Rearrangements so Frequent in Beetles? A Study of 50 Cases

**DOI:** 10.3390/genes14010150

**Published:** 2023-01-05

**Authors:** Bernard Dutrillaux, Anne-Marie Dutrillaux

**Affiliations:** Institut de Systématique, Évolution, Biodiversité, ISYEB-UMR 7205–CNRS, MNHN, EPHE, Sorbonne Université, 57 rue Cuvier CP50 F, 75005 Paris, France

**Keywords:** Polyphagan Coleoptera, X chromosome, rearrangements, insertion, Y-loss, evolution

## Abstract

Amongst the 460 karyotypes of Polyphagan Coleoptera that we studied, 50 (10.8%) were carriers of an X autosome rearrangement. In addition to mitotic metaphase analysis, the correct diagnosis was performed on meiotic cells, principally at the pachytene stage. The percentages of these inter-chromosomal rearrangements, principally fusions, varied in relation to the total diploid number of chromosomes: high (51%) below 19, null at 19, low (2.7%) at 20 (the ancestral and modal number), and slightly increasing from 7.1% to 16.7% from 22 to above 30. The involvement of the X in chromosome fusions appears to be more than seven-fold higher than expected for the average of the autosomes. Examples of karyotypes with X autosome rearrangements are shown, including insertion of the whole X in the autosome (ins(A;X)), which has never been reported before in animals. End-to-end fusions (Robertsonian translocations, terminal rearrangements, and pseudo-dicentrics) are the most frequent types of X autosome rearrangements. As in the 34 species with a 19,X formula, there was no trace of the Y chromosome in the 50 karyotypes with an X autosome rearrangement, which demonstrates the dispensability of this chromosome. In most instances, C-banded heterochromatin was present at the X autosome junction, which suggests that it insulates the gonosome from the autosome portions, whose genes are subjected to different levels of expression. Finally, it is proposed that the very preferential involvement of the X in inter-chromosome rearrangements is explained by: (1) the frequent acrocentric morphology of the X, thus the terminal position of constitutive heterochromatin, which can insulate the attached gonosomal and autosomal components; (2) the dispensability of the Y chromosome, which considerably minimizes the deleterious consequences of the heterozygous status in male meiosis, (3) following the rapid loss of the useless Y chromosome, the correct segregation of the X autosome–autosome trivalent, which ipso facto is ensured by a chiasma in its autosomal portion.

## 1. Introduction

Rearrangements involving both autosomes and sex chromosomes are infrequent in most animals. As for other chromosome rearrangements and genome mutations in general, the probability of their fixation, during evolution, depends on their occurrence (mutation rate) and the capacity of their maintenance along generations (resistance to selection pressure). As noticed in insects [1] and primates [2], the distribution of the types of chromosome rearrangements, which have accumulated in different lineages, is not at random. For instance, during primate evolution, the karyotypes of great apes and man principally differed by inversions, those of Cercopithecini by fissions, and those of lemurs by Robertsonian translocations. This led to the concepts of orthoselection [1] and/or orthomutation [2]. An important particularity of X autosome rearrangements (rea(X;A)) is related to the differences in the control of gene expression between sex chromosomes and autosomes, in both somatic and germ cells from males and females. In somatic cells of mammals, the concept of the gene dosage compensation between XX females and XY males, principally achieved by the random inactivation of one X in females [3], remains globally correct, although there are gene-to-gene variations of expression, with a proportion of genes escaping inactivation [4]. The inactivation of the X is driven by the inactivation center (Xic), which contains the Xist gene, which is able to mediate the silencing of the adjacent chromatin [5]. This inactivation of the X is always associated with its late replication. A major consequence of X autosome rearrangements in mammalian female somatic cells is the displacement of a portion of the X far from the Xic, which induces its reactivation, as well as the inactivation of the exchanged autosomal fragment placed under the Xic’s control. The resulting partial functional disomy X and autosomal monosomy may deeply alter development and cause malformations [6,7,8]. In germ cells at the pachytene stage of meiosis, the situation is different: while in females, both Xs are transcriptionally active, in males, the X and Y are embedded in the H2aX histone [9], form the sex body, and are inactivated. In the case of rea(X;A), the displacement of the autosomal fragment linked to the X in the sex body may cause its inactivation, which may alter the end of the meiotic process and impair fertility. This explains why, in our species, X autosome translocations are principally ascertained through dysmorphias and intellectual disability (somatic effect) in women and sterility (germinal effect) in men [6]. These detrimental consequences of rea(X;A), added to the risk of chromosome imbalance inherent in any type of chromosome rearrangements, should prevent the occurrence and transmission of X autosome rearrangements, thus their fixation during evolution. However, such rearrangements occasionally occurred and were fixed during mammalian evolution, but they did not occur at random. They were restricted to some lineages, mainly amongst Gerbillinae (Rodentia) [10,11,12] and gazelles (Bovidae, Antilopinae) [13]. In the species involved, females are homogametic XX (in fact neoXneoX) and males have complex sex formulas such as X1Y1Y2 or X1X2Y1Y2. An important point is how the inactivation spreading effect due to the X autosome fusion can be overcome, both in female somatic cells and in male meiotic cells. Early reports on human cytogenetics provided a first answer: the intercalation of heterochromatin between the autosomal and the gonosomal fragment of the neoX seems to prevent the inactivation spreading [14]. This notion was strongly supported by the fact that, in mammals, with a rea(X;A), heterochromatin is frequently present at the X autosome junction [15]. Hence, a condition facilitating these translocations is the acrocentric form of the X, as in Gerbillinae, which can place the X centromeric heterochromatin between the whole X and any attached autosome fragment on the X short arm. Heterochromatin seams to play the same role in male germinal cells during pachytene, a stage when many genes are expressed. In Gerbillinae, the heterochromatin, intercalated between the autosomal and gonosomal fragments, is at an external position of the sex body (purely gonosomal), and the attached autosomal fragment is projected outside, escaping inactivation [16]. These first observations on Gerbilles were confirmed in gazelles [13], but some examples without heterochromatin intercalation were described [12], which suggests that some non-heterochromatic chromosome components are able to stop the spreading of the Xist (see below).

In most insects, and beetles in particular, the sex chromosome system recalls that of mammals with XX females and XY males. However, the regulation of the gene dosage between these two sex formulae is different. In female somatic cells of genus Crioceris (Coleoptera, Chrysomelidae), both Xs remain euchromatic and replicate before heterochromatin [17]. This confirms the lack of X-inactivation in females, as demonstrated by microarray-based studies on gene transcription in *Tribolium* (Coleoptera, Tenebrionidae) [18]. These last studies also indicated the hyper-expression of the Xs in both sexes: the average expression of the X-linked genes in XX females is slightly higher than that of autosomal genes, whereas the single copies of the X-linked genes of XY males have the same level of expression as that of autosomal genes in two copies. Thus, the gene dosage compensation between males and females is principally achieved by the over-expression of the unique X of the males. This conclusion, close to that obtained in *Drosophila* (Diptera) [19], may prevail for most, if not all insects. Obviously, the lack of inactivation of the X in XX female beetles implies the lack of an ortholog of the Xist gene, but how the somatic (over-)expression in the male of the X is regulated remains unknown. This raises the question of the possible over-expression of the autosomal fragment linked to the X in the case of translocation, especially in male somatic cells. The problem is different for germ cells in male beetles. At pachynema, the sex chromosomes are strongly heterochromatized and embedded in argyrophilic (nucleolar) proteins [20]. We repeatedly confirmed this finding, in both XY and XO males [21], but in all the specimens carrying a rea(X;A) that we studied, while the X fragment was heterochromatized, the attached autosomal fragment had the same appearance as free autosomes and, thus, seemed to escape inactivation [22]. In most cases, C-banded heterochromatin was found intercalated between the autosomal and gonosomal fragments, suggesting its plug effect, being able to stop a cis-acting message, as for the Xist of mammals. Finally, in spite of many uncertainties about the mechanisms involved, it is clear that, in insects as in mammals, the transmission of chromosome rearrangements imposes more constraints when gonosomes rather than only autosomes are involved. This paper aimed to: (1) adapt for insects the international nomenclature for chromosome rearrangements, initially proposed for man and other mammals [23]; (2) show that, in beetles, the X is the chromosome the most frequently involved in inter-chromosomal rearrangements in spite that it may be deleterious, following a potential position effect; (3) explain the reasons for this paradox.

## 2. Materials and Methods

The techniques used were described in detail in [23]. They included sequential Giemsa staining, C-banding, and silver staining and were applied to establish both the mitotic karyotype of more than 500 species of Coleoptera and, in most instances, to analyze meiotic cells from pachynema to Metaphase II stages. No gonosome–autosome rearrangement could be demonstrated in Adephagan beetles, although they generally possess easily identifiable large gonosomes. Thus, we will focus on the Polyphaga suborder, of which 460 species were studied. An X autosome rearrangement was detected in 50 species, which are ranked by families, sub-families, and eventually, tribes in Table 1.

Table 1 lists the species with an X autosome rearrangement (rea(X;A)) amongst 460 karyotyped Polyphagan beetles. The abridged formulae indicate the types of rearrangements (see the Materials and Methods Section). The structures at the X autosome junction are as follows: c = centromeric heterochromatin; ter = terminal (telomeric) region, int = interstitial; h = intercalary heterochromatin, NOR = nucleolus organizer region or/and silver stained. References correspond to our previous publications on X autosome translocations; all other cases were unpublished.

Nomenclature: In absence of an official cytogenetic nomenclature for beetles, rearrangements involving the X chromosome, when detected, have been variously annotated in the literature, but derivative chromosomes are generally referred to as neoX and neoY. Amplifications of heterochromatin and the addition of autosomal material have not always been distinguished, creating some confusion. In this work, we will try to compromise between the previous literature on beetles and the principles of the international nomenclature on chromosomes adopted for mammals [31]. The reconstruction of the rearrangements originating the so-called neo-sex chromosomes shows that almost all are in fact whole-chromosome (X autosome) fusions associated with Y chromosome loss in the male (see below). Therefore, the so-called neoY is always the remaining free autosome. We will continue to use neoX and neoY, when the rearrangement is not specified, but additional abbreviations (Table 1 and Table 2) are proposed (t = translocation; q = long arm; p = short arm; A = Autosome; c = centromere; psu dic = pseudo-dicentric; rea = rearrangement; ter = terminal; add = addition; H = heterochromatin; der = derivative chromosome; ins = insertion; NOR = nucleolus organizer region [31].

## 3. Results and Discussion

The application of the ISCN nomenclature [31] to beetle karyotyping allowed us to define the rearrangements observed in this study. For comparison, a 20,XX karyotype, with free sex chromosomes, is shown in Figure 1.

- t(XqAq) = X autosome translocation, probably resulting from a Robertsonian translocation (or whole-arm fusion). The neoX is formed by the fusion, at the centromere regions, of the original X and an autosome, both acrocentric. Thus, the X and autosomal materials are separated by juxta-centromeric heterochromatin. The neoY is in fact the remaining unpaired acrocentric autosome. The developed formula should be, as for the following example of *Dynastes tityus* (Figure 1): 18,-X,-Y, -A,+der t(XqAq).

- ter rea(A;X) = end-to-end fusion between the X and a non-acrocentric autosome. No C-banded heterochromatin is visible at the presumed X/autosome junction. Example: *Sagra laticollis*: 22,-X,-Y,-A,+ter rea(A;X)(qter; p or q ter). At pachynema, the heterochromatized X is in the terminal position (Figure 2). The active centromere is that of the autosome. The NOR forms the chromosome 9 short arm

- psu dic(A;X) = as above, but heterochromatin is present at the X/autosome junction. The active centromere is that of the autosome. In *Coelosis biloba* (Figure 3, left): 18,-X,-Y,-A,+psu dic(A;X) (p;p), the interstitial C-band presumably corresponds to the juxta-centromeric heterochromatin of the original X. The neoY is an unpaired acrocentric autosome. As in several other species, the nucleolus organizer (NOR), embedded in the C-banded heterochromatin, is intercalated between the X and the autosomal portions. In *Stenopterus rufus* (Figure 3, right): 12,-X,-Y,-A,+psu dic(X;A)(p;q), heterochromatin is inconsistently C-banded, as in many other species of Cerambycidae, Its presence is marked by the coalescence of sister chromatids, which appears as its secondary constriction (Arrow head). The neoY is a large sub-metacentric autosome.

- ins(A;X) = insertion of the X into an autosome that kept its active centromere, while that of the X looks inactivated: 16,-X,-Y,- A,+der(A)ins(A;X). Examples: *M. planata* and *A. elongata* (Figure 4), which have the same mitotic karyotype. The presence of neo-sex chromosomes was known in these species [32], but not the diagnosis of insertion.

- complex sex = multiple sex and generally the presence of B chromosomes, as in genera *Blaps* and *Gnaptor* (Tenebrionidae) [22]. This category corresponds to the few cases of rearrangements too complex to be completely reconstructed by classical cytogenetic methods.

- neoXY = unspecified X autosome rearrangement. The neoX generally corresponds to the chromosome derived from a fusion X autosome and the neoY to the remaining free autosome. Other examples of X autosome rearrangements are shown in [22].

Diagnostic tools for X autosome rearrangements: Due to the lack of tools for indisputable identification of mitotic chromosomes in beetles, the diagnosis of rearrangements involving gonosomes was not easy in heterogametic males and almost impossible in homogametic females. The best criterion is the replacement of the sex bivalent by a more complex figure, generally a trivalent, at the pachytene and Metaphase I stages of male meiosis. At pachynema, in the absence of X autosome rearrangement, synapsed autosomes form fairly identifiable bivalents and sex chromosomes are usually totally heterochromatized, embedded in nucleolar proteins. At Metaphase I, sex chromosomes form the so-called parachute bivalent Xyp [20]. In the case of X autosome rearrangement, the neo-sex chromosomes form a more complex figure at pachynema, with a fragment similar to free autosomes, representing the autosomal portion in normal synapsis, prolonged by a heterochromatized fragment, representing the gonosomal portion (Figure 2). Not only this displays the rearrangement, but it also indicates that the attached euchromatic autosomal fragment escapes heterochromatization, hence inactivation. Indeed, this particularity of the meiotic prophase is associated with: (1) in mitotic cells, an even number of chromosomes, lower than that of closely related species with XY (Xyp) sex chromosomes, which is explained by the lack of the punctiform Y chromosome and the presence of an unpaired autosome (the so-called neoY); (2) at meiotic Metaphase I, the replacement of the parachute Xyp by an asymmetrical trivalent composed of the normally synapsed autosomal and the heterochromatized X portions; (3) at Metaphase II of meiosis, the presence on an asymmetrical chromosome (chimera half neoX/half neoY) as a result of a crossing over in the autosomal portion at the preceding Metaphase I (Figure 5). Insertions of the X in an autosome are visible at pachynema: the heterochromatized X is not in a totally terminal position and thrown out of the synapsed autosomal portion, as in *A. elongata* (Figure 4b; see below).

### 3.1. Relationships between X Autosome Rearrangements and Chromosome Number Variation

X autosome rearrangements were observed in at least one species from 14 different families, which represented 448/460 Polyphagan species studied. The diploid chromosome numbers ranged from 12 to 55 with a clear mode at 20 (221/460 = 48% of species; Figure 6). This modal number is that of the presumed ancestral number [20, 21], largely conserved here. Amongst these 221 species, only 6 (2.7%) had an X autosome rearrangement.

The decrease of the chromosome numbers can be seen in Figure 6 and Table 2.

Amongst the 460 species of Polyphagan beetles of this study, the percentages of karyotypes with a number of chromosomes above, below, or equal to 20 were 32, 20, and 48, respectively.

Amongst the 93 species with chromosome numbers less than 20, 34 species (36%) have a 19,X male karyotype. Most X0 males belong to the following families, sub-families, or tribes: Cantharidae; Lampyridae; Elaterinae and Lepturinae (Lepturini) in which the Y chromosome was lost early during their evolution [20,33,34,35]. In none of these karyotypes did the X chromosome look rearranged.Only 3/59 species with less than 19 chromosomes have an odd chromosome number (2n = 15 or 17): their X is apparently normal, and their Y is lost. In species with less than 19 chromosomes and even chromosome numbers, the occurrence of X autosome rearrangements is particularly high, i.e., 30/56 = 53.6%.On the whole, 32.2% (30/93) of karyotypes with less than 20 chromosomes display a rea(X;A).

The increase of chromosome numbers can be seen ion Table 2 and Figure 6:Chromosome numbers’ increases are more frequent than their decreases: 146 versus 93, and most of them (92/146 = 63%) correspond to karyotypes with 22 or 24 chromosomes.In taxa in which some species had an increased chromosome number, principally Cerambycinae, Lamiinae, Prioninae, Chrysomelinae, Galerucinae, and Curculionidae, many other species conserved the presumed ancestral formula 20,XY. Thus, the number increases were secondary events. In these species with increased chromosome numbers, rearrangements of the X chromosomes are not exceptional and the increases parallel those of the total number of chromosomes as a witness to an active chromosome evolution. Nevertheless, they remain much less frequent than in species with low chromosome numbers. On the whole, 14/146 (9,6%) karyotypes with more than 20 chromosomes display an X autosome rearrangement.

### 3.2. Chromosome Fusions Preferentially Involve the X

With the limited tools for beetle chromosome identification, reciprocal autosome exchanges are almost impossible to detect with certitude. Only chromosome number variations provide indisputable criteria to track autosome rearrangements such as fissions and fusions. Thus, most of the 2n = 18 formula were assumed to derive from the ancestral 2n = 20 by one homozygous fusion. If gonosomes and any autosome pair from the 20,XX/20XY population were equally involved in fusions, the X should be involved only 3/40 times (1/10 in females and 1/20 in males: *p* = 0.07). That neo-sex chromosomes were observed in about half of the karyotypes with 18 chromosomes, demonstrates the very preferential involvement of the X in fusions (0.50 instead of 0.07 = 7.1-times more than expected in the case of a chromosome involvement at random). The behavior of the punctiform Y chromosome will be discussed below.

### 3.3. Relationships between X Autosome Rearrangements and Chromosome X Morphology

As evidenced by the high occurrence of Robertsonian translocations in animals, acrocentric chromosomes are more frequently involved in fusions than metacentric chromosomes. In most families of beetles, the basic 20,XY karyotype is essentially composed of metacentric and sub-metacentric autosomes, which obviously does not favor the occurrence of a Robertsonian evolution. However, different from autosomes, the X chromosomes look acrocentric in many species. Furthermore, our systematic use of C-banding and silver staining shows that the short arm of a large proportion of apparently non-acrocentric Xs is only composed of heterochromatin, which often harbors the NOR [34]. We propose that the NOR behaves like a fragile site and is the site of recurrent rearrangements. These rearrangements remain at the cellular level when the NORs are interstitial (they generate imbalances at meiosis and are eliminated), but they can pass to the next generations when they are in the terminal position [36,37]. Thus the X, frequently acrocentric and occasionally the NOR carrier on its heterochromatic short arm, is more prone than metacentric autosomes to undergo transmissible rearrangements, end-to-end fusions in particular.

In Polyphagan Coleoptera, whereas most often autosomes have a metacentric or sub-metacentric morphology, the X chromosome more frequently looks acrocentric (in about 30% of the species we studied), and when it looks non-acrocentric, its short arm is frequently composed of C-banded heterochromatin, which occasionally harbors the NOR. This condition should facilitate translocations of autosomal components on the short arm because (1) the NOR behaves like a fragile site, which promotes exchanges [26,37], and (2) the centromeric heterochromatin would insulate the X proper component and any attached chromosome fragment (see below). Thus, the occurrence of rea(X:A) should be expected to be higher in taxa in which the X is frequently acrocentric. Our sample (Table 2) is not sufficient to confirm this hypothesis, but it is noticeable that the rea(X;A)s are rare in Cerambycidae (5/121= 4%), whose X chromosome is frequently non-acrocentric, and more frequent in Dynastinae (13/49 = 26%) or Lucanidae (4/14 = 29%), whose X chromosome is frequently acrocentric. However, the small number of rea(X;A)s in Cetoniinae (1/63), in which the X chromosome is frequently acrocentric, relativizes the validity of this correlation.

### 3.4. X Chromosome Insertion into Autosome

With the lack of R- or G-banding and chromosome painting in beetles, complex chromosome rearrangements, such as insertions, are difficult to diagnose. We saw above that, in the case of X autosome translocation, the distally attached sex body, composed of the X alone, forms a trivalent with the synapsed autosomes (Figure 2). Insertions of DNA sequences constitute a common type of mutation, known in Drosophila, as well as in human pathology [38,39], but to the best of our knowledge, whole chromosome insertions were never described. We show here that not only this occurs, but also that it can be maintained during evolution. After insertion of an autosomal fragment, meiotic recombination leads to highly deleterious duplications/deficiencies, which are rapidly eliminated. Here, the inserted X is heterochromatized and thrown out of the synapsed autosomal portion, as in *A. elongata* (Figure 4). Thus, crossing over occurrence, strictly limited to the normally synapsed autosomal portion, allows the correct chromosome segregation and does not generate imbalances (see below). As shown in *C. elegans,* the rearrangement, however, occasionally may induce a partial asynapsis of the adjacent autosomal portion, but the rarity of the event (in our experience, in about 1/100 spermatocytes I; Figure 7) should not impair gametogenesis.

The long-term transmissibility of the X chromosome insertion is demonstrated by the presence of the same acrocentric neoX in *A. elongata* and *M. planata* (Tenebrionidae, Pimelinae, Akidini), two species from Spain and Morocco, respectively, which have a fairly similar 2n = 16 karyotype [32]. The 2n = 22 karyotype of *C. elegans* (Tenebrionidae, Tenebrioninae, Scaurini) also displays an ins(A;X) (Figure 7), but the neoX is sub-metacentric and both the neoY and neoX larger than those of the other two species. Thus, either the original insertions involved different autosomes or a single insertion occurred, but the autosomal part of the neo-sex chromosomes was further rearranged. If the initial rearrangements were different, the occurrence of a rare chromosome rearrangement in three related species would suggest that it was facilitated by a particular chromosome structure.

### 3.5. Heterochromatin Frequently Insulates the X Fragment after X Autosome Fusion

As recalled above for mammalian somatic cells, heterochromatin intercalation between the attached X and autosome can modulate the consequences of X autosome fusions, by preventing the diffusion of a message (Xist RNA) from the inactive X to the autosomal part [5,6]. However, this “plug effect” may be limited to the spreading of the Xist RNA from the mammalian late-replicating X of females and, thus, be irrelevant in insects both Xs of which are early-replicating in female cells [17]. However, the high level of expression of X-linked genes compared to autosomal genes, in both male and female insects [18], strongly suggests the existence of a cis-acting message driving gene hyper-expression, but it is not known if this hypothetical message can spread on the attached autosomal fragment in the case of X autosome translocation. Only indirect arguments can be proposed, such as the recurrent presence of heterochromatin at the X/autosome junction, which might stop the message, by analogy with mammalian chromosomes. Indeed, C-banded heterochromatin is present at the X-A junction in most instances with a reasonable interpretation of the rearrangement (Table 1):A t(XqAq) was found in 26 species. In somatic cells, the neoY (the free autosome) was acrocentric and the peri-centromeric heterochromatin, which separates the X proper and the autosomal fragment, was often large and always larger than that of the free autosomes (Figure 1). In germ cells at pachynema, the sex trivalent was composed of the normally synapsed autosomes prolonged by a heterochromatized fragment;A psu dic (X;A) was found in four species, as indicated by the presence of an interstitial C-band (Figure 3);A ter rea(X;A) was found in 13 species. The neoX was as above, but the neoY was sub-metacentric. Thus, either a pericentric inversion involved the neoY, or a termino-terminal fusion occurred, with inactivation of one centromere. In that case, telomeric repetitive sequences may insulate the X and the autosome;In five instances, amplified C-banded heterochromatin was dispersed on the X proper fragment.

On the whole, C-banded heterochromatin could not be evidenced at the X autosome junction in 15/50 species, but the example of *P. didymus* shows that other heterochromatic components may replace it. In this species (Figure 8), as in many other beetles, centromere regions are both silver-stained and C-banded. In some taxa, as in many Melolonthinae, however, centromere regions may be stained by silver only, which indicates that silver may stain a protein content of non-C-banded heterochromatin. In *P. didymus*, silver stains centromere regions, as C-banding does, but in addition, a chromatin fragment of the neoX at the presumed X autosome junction, which evidences the presence of non-C-banded heterochromatin in which the NOR is nested (Figure 8). Thus, by analogy with rearrangements of the X in mammals [6,14,16], these observations suggest that, in the case of X chromosome translocations in beetles, particular chromatin constitutions, most frequently C-banded heterochromatin, insulate the X proper from adjacent autosomal components. In corollary, this may indirectly indicate the existence of cis-acting regulatory elements, which either induce somatic hyper-expression (in both sexes) or inactivation of the X in the meiotic prophase of the male. Another no-less-important plug effect of heterochromatin exists in male germ cells at pachynema, a phase when autosomes, but not gonosomes, are highly transcribed. In t(X;A) carrying mammalian species, we saw above that heterochromatin could maintain the autosomal fragment out of the inactivated sex body, preventing its inactivation and, consequently, a probable decrease of fertility [16]. In beetles, there is no proper sex body at pachynema, but the sex chromosomes, embedded in argyrophilic (nucleolar) proteins, are very probably inactivated. In the case of X-A fusion, the autosomal portion is synapsed similarly to free autosome bivalents, but prolonged by the heteropycnotic gonosomal fragment (Figure 2).

### 3.6. Y Chromosome Loss and the Nature of the neoYs

As proposed [20], the Y chromosome may have only a mechanical role in beetles for the correct segregation of the achiasmatic X at Anaphase I of male meiosis. This interpretation is confirmed by microarray studies [18], which did not detect any transcript from the Y chromosome of the flour beetle, suggesting it is devoid of genes. Among the 460 sex formulae of this study, 52 (11.3%) are of the X0 type including 34 (7.4%) with a 19,X formula (Figure 6). This X0 formula, associated with an uneven number of chromosomes, may be present in a whole family (Elateridae, Lampyridae) (20), tribe (lepturini: Cerambycidae) [35], or genus (*Odontolabis*: Lucanidae) [43]. It may be also present in a single or a few species, such as *Protaetia vidua* Gory & Percheron 1833 within the large *Protaetia* genus, which usually has a 20,XY formula (personal data). Thus, the simple loss of the Y independently recurrently occurred at diverse stages of evolution, and this does not seem to correlate with the acquisition of any phenotypic character. In X0 male spermatocytes, the X remains embedded in nucleolar proteins until Anaphase I, which ensures its correct segregation [21]. In beetles with neo-sex chromosomes, there are no traces of the Y, neither as a free minute chromosome nor in the neoY. There is no more equivalent of the Xyp at Metaphase I, and a chiasma in the autosomal fragment of the neoXY bivalent ensures the correct segregation of the neoX and neoY at Anaphase I. Thus, we posit that gonosome autosome rearrangements (1) involve only the X and an autosome and (2) systematically lead to the loss of the useless Y. Consequently, the so-called neoY is purely autosomal. This interpretation differs from that proposed about the composition of the neoX and neoY following an X autosome translocation in *Timarcha aurichalcea* (Tenebrionidae) [40], in which the conservation of the Y and the partial deletion of the X were privileged.

### 3.7. The Dispensability of the Y Facilitates the Transmission of the neoX Chromosome

In most animals, either of the XX/XY or ZZ/ZW types, the gene content of the sex chromosomes is very unequal: rich in the X and Z and poor in the Y and W; however, even the Y and W chromosomes contain indispensable genes and their lacking is either lethal or leads to sterility. Thus, in most species with gonosome–autosome rearrangements, the two gonosomes are involved, which ensures their correct meiotic segregation, as in Gerbilles and Gazelles [13,16]: only a few species with a t(Y;autosome), but none with females heterozygous for a t(X;autosome) are known. The situation is different in beetles, in which only the X is essential. The only role of the Y chromosome is to contribute to the formation of the asynaptic parachute sex bivalent, assumed to facilitate the correct segregation of the X at Anaphase I of meiosis. However, this system is faulty and leads not only to recurrent losses of the Y, but also to the high incidence of XYY males [41]. X autosome fusions allow the remaining free autosome and the autosomal part of the neoX to exchange at pachynema, which ipso facto ensures the chiasma formation at Metaphase I and their correct meiotic segregation at Anaphase I. Thus, following the chromosome rearrangement, the delicate period of heterozygosity is largely spared and the useless Y can be rapidly lost without any deleterious consequence.

The question remains of the competition between X autosome translocation carriers and non-carriers. The reproductive fitness depends on the regularity of chromosome segregation at meiosis. In non-carriers, the sex chromosome segregation ensured by the parachute system is faulty, as shown by the high incidence of XYY, XYYY, and X0 males [41]. Y polysomy leads to a decrease of fertility, as shown for triplo Y males in genus *Dermestes* [42]. Thus, the meiosis of the chiasmatic neoXY males appears to be less error-prone than that of Xyp males, which may favor the spreading of the neo-sex chromosome carriers amongst Xyp populations.

## 4. Conclusions

The gene dosage compensation between the XY or X0 male and XX female beetles is principally achieved by the over-expression of the genes of the unique X of the males different from the inactivation of one X in mammalian females. Because of a possible spreading effect on the autosome of regulatory elements (such as the Xist in mammals), this should limit the involvement of the X in rearrangements with autosomes. However, paradoxically, during beetle evolution, the X chromosome appears to be involved much more frequently than the average autosomes in inter-chromosomal rearrangements. It is proposed that this paradox is principally the indirect consequence of the dispensability of the Y chromosome. After X autosome translocation, the correct segregation at meiosis of the rearranged X autosomes trivalent is ensured by a chiasma in its autosomal part, whereas the free Y, prone to segregation errors, is rapidly lost without detriment. Consequently, the X autosomes trivalent behaves like a bivalent as regards meiotic segregation errors, and the heterozygous status is much less risky for progeny than that for any autosome-to-autosome rearrangement. This emphasizes the predominant role of the selective pressure against the heterozygous status in the limitation of chromosome rearrangements during evolution. This limited effect of selection against X autosome rearrangements also explains their high and fairly independent recurrence during evolution, as shown for the Lucanidae family [43].

## Figures and Tables

**Figure 1 genes-14-00150-f001:**
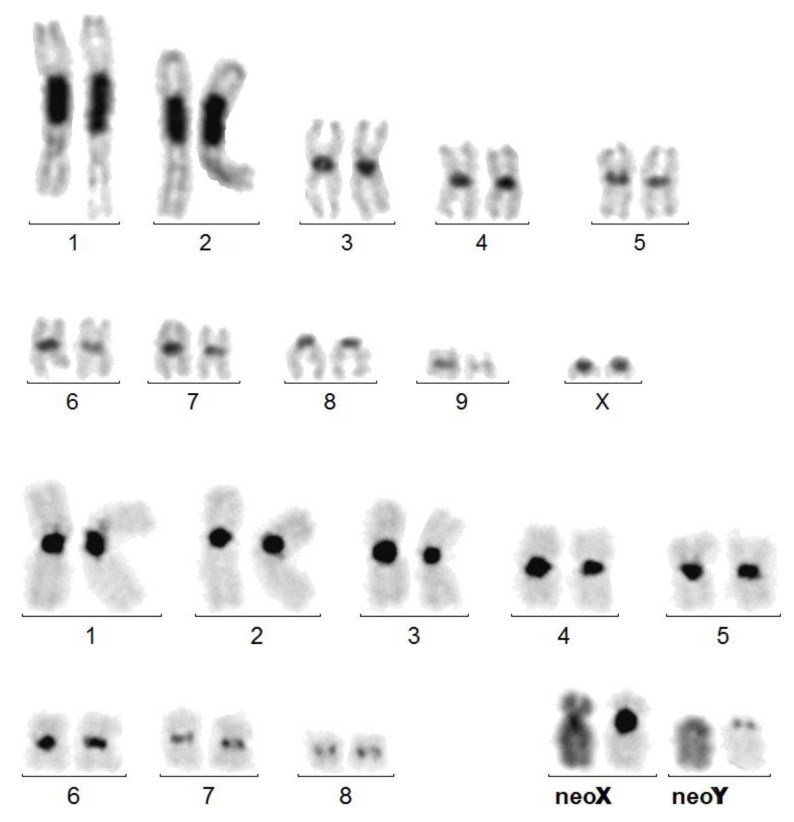
Upper karyotype: *Dynastes neptunus*. 20,XX female C-banded karyotype, close to that of the ancestor, except for the amplification of heterochromatin (dark, chromosomes 1 and 2) and the acrocentric, instead of metacentric form of chromosome 8. Lower karyotype: *Dynastes tityus.* C-banded karyotype exhibiting a Robertsonian translocation: t(XqAq). The neoY and the Xq arm are purely autosomal. As in other cases of t(XqAq), the C-band at the X autosome junction on the neoX is much larger than that on the neoY. Giemsa-stained neoX and neoY have been added. The neoY is presumably the ortholog of chromosome 8 of *D. neptulus*.

**Figure 2 genes-14-00150-f002:**
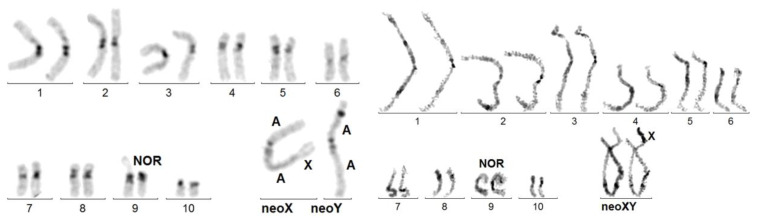
*Sagra laticollis.* Left figure: C-banded mitotic karyotype exhibiting a ter rea(X;A). The dark band on the neoY short arm is an artifact. Right figure: sequentially Giemsa-stained (left bivalent) and C-banded (right bivalent) karyotype of a spermatocyte at pachynema: the X component on the neoXY bivalent is terminal and heterochromatized.

**Figure 3 genes-14-00150-f003:**
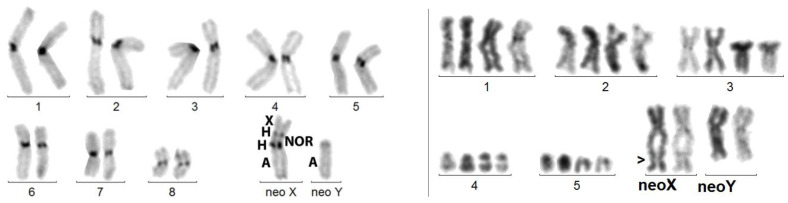
**Left** figure: *Coelosis billoba.* C-banded karyotype exhibiting a psu dic (X;A). The original X, probably sub-metacentric, carried the NOR on its short arm. Its centromere was inactivated. **Right** figure: sequentially Giemsa-stained and C-banded karyotype of *Stenopterus rufus*. The arrowhead points to the secondary constriction at the place of the (inactivated?) centromere of the X proper. The fusion presumably involved the telomere regions of the sub-metacentric autosome long arm and acrocentric X short arm.

**Figure 4 genes-14-00150-f004:**
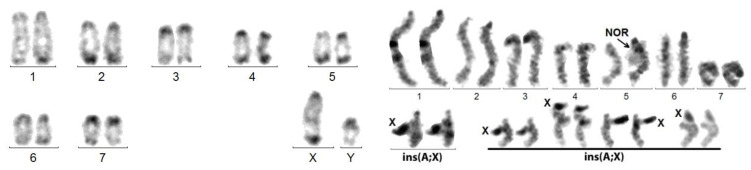
Left figure: C-banded mitotic karyotype of *M. planata*. X and Y are in fact neo-chromosomes. Right figure: sequentially Giemsa-stained (left bivalent) and silver-stained (right bivalent) karyotype of a spermatocyte I at pachynema of *A. elongata.* An additional series of 4 sex trivalents exhibiting the ins(A;X) from 4 other cells shows that the heterochromatized X component is recurrently thrown aside, whereas the autosomal component is normally synapsed. This original configuration is explained by the insertion of the original X into an autosome.

**Figure 5 genes-14-00150-f005:**
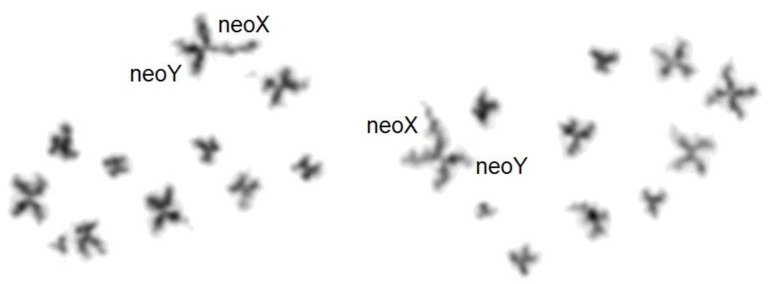
Two brother spermatocytes at Metaphase II of *S. laticollis* exhibiting identical chimeric sex chromosomes following a crossing over in the autosomal portion of the neo-sex trivalent, which will give the so-called neoX and neoY.

**Figure 6 genes-14-00150-f006:**
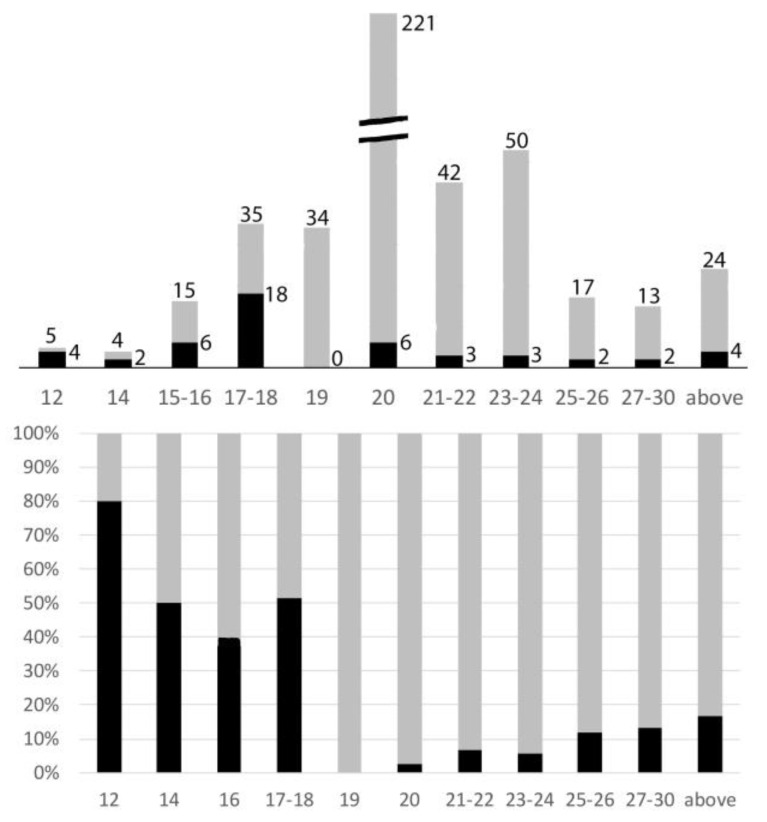
Relationships of rea(X;A) versus chromosome numbers. **Upper** figure: histogram indicating the distribution of chromosome numbers amongst the species studied (numbers at the top). The numbers of karyotypes with an X autosome rearrangement (black) is indicated on the right side. **Lower** figure: same data giving the percentages (black) of karyotypes with an X autosome rearrangement in relation to their total chromosome numbers.

**Figure 7 genes-14-00150-f007:**
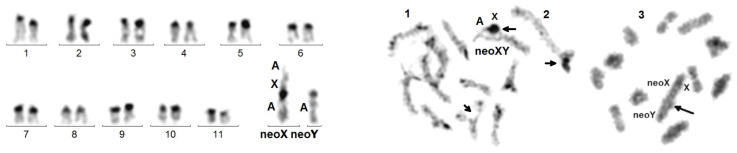
*C. elegans*. Left figure: C-banded mitotic karyotype composed of acrocentric chromosomes, excepting neoX. Right figure: meiotic cells: (1) late pachynema with partial de-synapsis (small arrows) of the NOR (autosomal, bottom) and of the X adjacent regions of the neoXY trivalent (top); (2) usual morphology of the neoXY trivalent (isolated from another cell); (3) diakinesis showing the distal chiasma (arrow) linking the autosomal portions of the neoX and neoY.

**Figure 8 genes-14-00150-f008:**
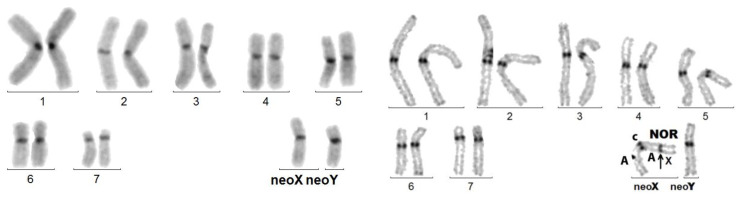
Karyotypes of *P. didymus*. **Right** figure: C-banding. **Left** figure: silver staining exhibiting a heterochromatic component, i.e., an inactive centromere or the NOR, at the presumed X autosome junction. A misidentification is possible between Pair 6 and the neoY.

**Table 1 genes-14-00150-t001:** Species with a rea(X autosome).

Family: Sub-Family: Tribe	Species with a rea(X;Autosome)	Abridged Formulae	Junction	Ref.
Buprestidae: Chrysochroinae	*Capnodis tenebricosa* Olivier, 1790	14,psu dic(X;A) (p;p)	c-c	
Buprestidae: Julodinae	*Julodis ehrenbergi* Jaubert, 1858	26,ter rea(X;A)(p;p)	ter-ter	
Cantharidae: Cantharinae	*Cantharis sp* (Shar mt., Northern Macedonia)	12,t(XqAq)	c-c	
Cerambycidae: Cerambycinae	*Stenopterus rufus* L., 1767	12,psudic(X;A)(p;q)	ter-ter	
Cerambycidae: Lamiinae	*Dorcadion obenbergeri* Heyrovsky, 1940	22,t(XqAq)	c-c	
	*Dorcadion pseudolugubre* Breuning, 1943	22,t(XqAq)	c-c	
Cerambycidae: Lepturinae	*Desmocerus palliatus* Forster, 1771	23,complex	?	[24]
Cerambycidae: Prioninae	*Prinobius scutellaris* Germar, 1817	28,t(XqAq)	c-c	[25]
Chrysomelidae: Chrysomelinae	*Timarcha strangulata* Fairmaire 1861	20,t(XqAq),add H	c-h	
Chrysomelidae: Cassidinae	*Cassida rubiginosa* Müller, 1776	16,ter rea(X;A)(p;p)	ter-ter	
Chrysomelidae: Galerucinae	*Agelastica alni* L., 1758	24,t(XqAq)	c-c	
Chrysomelidae: Sagrinae	*Sagra femorata* Drury, 1767	20,ter rea(X;A)(p;q)	ter-ter	
	*Sagra laticollis* Fabricius, 1792	22,ter rea(X;A)(p;q)	ter-ter	
Coccinellidae: Coccinellinae	*Anatis ocellata* L., 1758	18,t(XqAq)	c-c	
Curculionoidae: Lixinae	*Cleonus piger* Scopoli, 1763	28,t(XqAq)	c-c	
Lucanidae: Lucaninae	*Cyclommatus metallifer* Boisduval, 1835	26,ter rea (X;A)(p;?)	ter-h	
	*Dorcus parallelepipedus* L., 758	18,t(XqAq)	c-c	[26]
	*Homoderus mellyi* Parry, 1862	12,ter rea(X;A)(p;?)	ter-ter	
Lucanidae: Lampriminae	*Phalacrognathus muelleri* Mcleay, 1885	16,t (XqAq)	c-c	
Melandryidae: Osphyinae	*Osphya bipunctata* Fabricius, 1775	18,ter rea(X;A)(p;?)	c-c	
Mordellidae: Mordellinae	*Variimorda villosa* Schrank, 1781	12,ter rea(X;A)(p;?)	?	
Pyrochroidae: Pyrochroinae	*Pyrochroa coccinea* L., 1761	18,t(XqAq)	c-h	
Scarabaeidae: Cetoniinae	*Jumnos ruckeri* Saunders, 1839	14,ins(A;X)	c-h	[27]
S.: Dynastinae: Dynastini	*Chalcosoma atlas* L., 1758	20,ter rea(X;A)(p;?)	ter-ter	[28]
	*Chalcosoma Caucasus* Fabricius, 1801	20,ter rea(X;A)(?;p)	ter-ter	[28]
	*Dynastes granti* Horn, 1870	18,t(XqAq)	c-NOR	[29]
	*Dynastes hercules Hercules* L., 1758	18,t(XqAq)	c-NOR	[29]
	*Dynastes h. reidi* Chalumeau, 1977	18,t(XqAq)	c-NOR	[29]
	*Dynastes tityus* L., 1758	18,t(XqAq)	c-NOR	[29]
S.: Dynastinae: Oryctini	*Augosoma centaurus* Fabricius, 1775	18,t(XqAq)	c-c	[28]
	*Coelosis biloba* L., 1767	18,psu dic(X;A)(p;p)	c-h	
	*Cyphonistes vallatus* Wiedermann, 1823	16,ter rea(X;A)(q;?)	?	
	*Oryctes nasicornis* L., 1758	18,t(XqAq)	c-c	[22]
	*Oryctes owariensis* Palisot de Beauvois, 1807	18,t(XqAq)	c-c	
	*Oryctes rhinoceros* L., 1758	18,t(XqAq)	c-c	[28]
S.: Dynastinae: Phileurini	*Phyleurus didymus* L., 1758	16,psu dic(X;A)(p;p)	ter-NOR	
S.; Melolonthinae	*Polyphylla fullo* L., 1758	18,neoXY	?	
S.: Trichinae	*Gnorimus variabilis* L., 1758	20,neoXY	?	
	*Osmoderma eremita* Scopoli, 1763	18,t(XqAq),+Bs	c-c	
	*Osmoderma lassallei* Baraud & Tausin, 1991	18,t(XqAq)	c-c	
	*Osmoderma scabra* P. de beauvois, 1805	18,ter rea(X;A)(p;p)+B	c-c	[22]
Silvanidae: Brontinae	*Uleiota planata *L., 1761	32,ter rea (X;A)(q;?)	?	
Tenebrionidae: Diaperinae	*Corticeus unicolor* Piller, Mitterpacher, 1783	18,t(XqAq)	c-c	
Tenebrionidae: Pimeliinae	*Akis elongata* Brullé, 1832	16,ins(A;X)	int/ter/ter	
	*Morica planata* Fabricius 1801	16,ins(A;X)	int/ter/ter	
Tenebrionidae: Tenebrioninae	*Blaps mucronata* Latreille, 1804	35,complex	?	
	*Cephalostenus elegans* Brullé, 1832	24,ins(A;X)	int/ter/ter	
	*Gnaptor spinimanus* Pallas, 1781	36,complex	?	[28]
	*Scaurus striatus* Fabricius, 1792	20,t(XqAq)	c-c	
Vesperidae: Vesperinae	*Vesperus xatarti* Mulsant, 1839	53,complex	?	[30]

**Table 2 genes-14-00150-t002:** Polyphagan beetle families, sub-families, or tribes in which we detected at least one rea(X;A). N1: number of species studied in each family. N2: number of species studied in each sub-family or tribe. n rea(X;A) and p rea(X;A): number and frequency of rea(X;A) in N2.

Family	N1	Sub-Family/Tribe	N2	n rea(X;A)	p rea(X;A)
Buprestidae	4	Chrysochroinae	1	1	1
		Julodinae	1	1	1
Cantharidae	2	Cantharinae	2	1	0.50
Cerambycidae	121	Cerambycinae	16	1	0.06
		Lamiinae	60	2	0.03
		Lepturinae	32	1	0.03
		Prioninae	10	1	0.10
Chrysomelidae	50	Cassidinae	2	1	0.50
		Chrysomelinae	19	1	0.05
		Galerucinae	12	1	0.08
		Sagrinae	2	2	1
Coccinelidae	7	Coccinellinae	7	1	0.14
Curculionidae	24	Lixinae	3	1	0.33
Lucanidae	14	Lampriminae	2	1	0.50
		Lucaninae	8	2	0.27
		Dorcinae	2	1	0.50
Melandryidae	1	Osphyinae	1	1	1
Mordelidae	1	Mordelinae	1	1	1
Pyrochroidae	2	Pyrochroinae	2	1	0.50
Scarabaeidae	164	Cetoniinae	63	1	0.02
		Dynastinae: Cyclocephalini	22	0	0
		Dynastinae: Dynastini	12	6	0.50
		Dynastinae: Oryctini	8	6	0.75
		Dynastinae: Phyleurini	2	1	0.50
		Other Dynastinae	5	0	0
		Melolonthinae	14	1	0.07
		Rutelinae	16	0	0
		Trichinae	9	4	0.44
Silvanidae	1	Brontinae	1	1	1
Tenebrionidae	31	Diaperinae	1	1	1
		Pimeliinae	13	2	0.15
		Tenebrioninae	12	4	0.33
Vesperidae	1	Vesperinae	1	1	1
Other families	37		37	0	0
**TOTAL**	**460**			**50**	**0.11**

## Data Availability

Complementary data can be found in our database http://insect-cytogenetics.fr/ (accessed on 10 November 2022).

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
