# Peer review of "Why Are X Autosome Rearrangements so Frequent in Beetles? A Study of 50 Cases"

_genes, 2023, doi:10.3390/genes14010150_

Round 1

Reviewer 1 Report

The MS is a large and insightfull study on 460 beetles of
suborder Polyphaga to find X-autosome rearrangements. I
suggest to give the number of checked beetles per family or
subfamily to know possible differences in their rea(X;A).
The updated number of known karyotyped species per taxa
should be given. Timarcha strangulata has not 2n=20 but 2n=
28 (Genet. iber. 22, 1970). Our FISH data on T. aurichalcea
chrom. origin (J.Zool.Syst.Evol.Res.42,2004) needs mention.

Author Response

I added a paragraph at the end of the discussion section, which includes one reference citation , and the reference was added in the references section

Reviewer 2 Report

This work presents interesting data regarding the involvement of chromosome X in different rearrangements which are stable in some species. Authors observed that, in Coleoptera, X chromosome is the one most frequently involved in inter-chromosomal rearrangements and authors aim to explain this observation.  However, the paper is not clear and well organized. Results are reported in a confused way, the description of results is partial and difficult to follow while methodological details are largely missing.

Here below, a list of all the points to be addressed.

MAJOR COMMENTS:

Introduction section:

-       authors refer to primates and other mammal species but, actually, it could be helpful to add something regarding the evolution or, rather, the phylogenetic relationship among the considered species (Coleoptera). They should also explain better the putative ancestral karyotype of these species. In addition, a brief description of some peculiar features of these species, such as centromere conformation and sex determination/gene dosage compensation, could be added.

Materials and Methods section:

-       the techniques are only listed and, even if a reference is cited, it should be better to explain some detail about the strategy of the work, the preparation of mitotic and meiotic chromosomes/bivalents. Authors should clearly describe how many species they analysed for the work (how many species, belonging to how many families) and maybe distinguish between karyotypes prepared for this paper and karyotypes already described;

-       since, if I understood correctly, Table 1 is the result of these staining and banding analyses, it could be inserted in the Result section. It could be simplified, leaving only Family or Tribe in the left column and only the name of the species in the second column. Authors may group the species on the bases of their rearrangement type or, rather, it should be helpful to add a different table summarizing the different classes of rearrangements.

-       the sentence “The reconstruction of…”, lines 139-142, could be moved in the Result section since it is describing an observation and not a method.

Results and discussion section:

-       in general, authors have to describe their data in a clearer and linear way. First paragraph reports some examples of rearrangement. Instead of that, authors have to describe the type of rearrangement, in how many species they found it without spreading details in different part of the text. For example, lines 349-361 should be included in this first paragraph;

-       it could be helpful an image representing a canonical karyotype, without rearrangement (X;A), to be compared with the rearranged ones;

-       it could be helpful to add a scheme/tree summarizing phylogenetic relationship among the evaluated species;

-       it is not clear to me what does it mean “neoXY” or “complex” referred to Polyphylla fullo, Gnorimus variabilis, Blaps mucronata and Gnaptor spinimanus in Table1. They are not described in the text;

-       the description of the classes of rearrangement is not clear in general, but, in particular at line 160, “fusion between the X and a non-acrocentric autosome”. Differently, in the associated figure 2, the autosomes are “at least two” (lines 166-167);

-       authors identify the active centromeres through staining analysis. Is there a different way to verify the position of the active centromere? For example, an immunofluorescence using an antibody against CENH3 may be used (Gržan T et al. PLoS Genet. 2020 Oct 30;16(10));

-       the paragraph “Chromosome fusion preferentially involve the X”, lines 260-270, is not clear and it should be extended with more details;

-       lines 391-392: it is an important to better explain this point;

-       it could be interesting to discuss the high frequency of these rearrangements with respect to the evolution of these species.

MINOR COMMENTS:

Authors should check very carefully the first part of Results and Discussion section in which many errors and inaccuracies are present. Some examples are listed below:

-       “an” instead of “and” at line 160;

-       figure legend and main text are not separated;

-       the size of the figures is not uniform.

Author Response

I changed many paragraphs and added two figures, as requested (Figures 1 left and figure 3 right). All changes in the draft are over lined in yellow in the copy joined. I hope they respond to most of the comments. Hovever, 1) I did not add the description of the techniques: they were largely and repeatedly described in our previous papers. 2) The number of species studied per family... was already given in table 2. 3) We did not performed immuno-fluorescence studies: I understand the point, but as retired scientist (no more grants), we had no mean to do it. Anyhow, the time to do it would be incompatible with the time required.

Reviewer 3 Report

Dosage compensation has been a topic of enigma across the animal kingdom for long. Depending upon the class and species, it may range from inactivation of one of the X chromosome in species with heteromorphic sex chromosomes (XX) or upregulation of the single X chromosome in the males (XY). The primary signals that initiate sex determination in insects are highly diverse. In the vinegar fruit fly Drosophila melanogaster, the double dosage of X-linked signal elements differentiates XX from XY embryos and triggers female development. However, this is not a uniform phenomenon across the insect kingdom. In many other insects, the presence or absence of a dominant male-determining factor determines the sex of the progeny. Information regarding X-autosome rearrangements over evolutionary period resulting in neo-sex chromosomes is sparse in beetles. Authors Dutrillaux and Dutrillaux provide a comprehensive study of Polyphagan coleoptera beetles to reveal the mechanism of dosage compensation in the species. Representative images depicting both Giemsa and C banding provide an easy to follow manuscript, despite the variety of results within the study.

Strengths- Excellent written article, comprehensive with images to follow for each section of results.

Detailed description and reasoning behind their hypotheses, regarding Robertsonian translocations involving X-autosomal chromosomes contrasting with the non involvement of Y chromosome

Suggestions – While not lacking in their reasoning and evidence behind the preferential involvement of X chromosome in XA rearrangements, I would like to see a discussion of what the potential driver mechanisms/system behind such a rearrangement could be in only a few rather than most Polyphagon species. At the same time, the manuscript stipulates the degeneration of neo-Y in such species due to the absence of any essential genes on the Y related to reproductive fitness. This leads to the question of the Y chromosome’s presence in the other coleopteran species. Does this stipulate that the species which do not have a Y degeneration or X-autosome rearrangement, there are essential genes benefial to the male sex located on the Y preventing their total loss? Have there been evidences of neo-X chromosomes reverting to ancient autosomal chromosomes? Additionally, in contrast to the evidence of selective pressure enforced through X-autosome recombination, another recent work has shown the role of sexually antagonistic impact of Y-linked inversions (Lenormand and Roze 2022). A short paragraph addressing this and comparing the sex chromosome evolution patterns in other species (eg. Bachtrog 2006, Vicoso and Bachtrog 2013) would add depth to this article’s discussion section.

Minor

1.     Line 77- heterochromatin ‘seams’ should be changed to ‘seems’

2.     Line 277-280 -Furthermore, our systematic use of C-banding and silver staining shows that the short arm of a large proportion of apparently non-acrocentric Xs is only composed of heterochromatin, which often harbors the NOR [34]. We proposed that the NOR behaves like a fragile site and is the site of recurrent rearrangements.

Figure 2 – C banding in Sagra laticollis shows NOR on chromosome 9. Does this suggest that chromosome 9 and X fused to form neo-X? Unclear.

Author Response

(The authors gave the same response as above.)

Round 2

Reviewer 2 Report

The paper has been improved and several points have been clarified. The paper can be accepted. The authors should check the space between paragrphs (starting from "X chromosome insertion into autosome").